# Structural Connectivity Alterations in Operculo-Insular Epilepsy

**DOI:** 10.3390/brainsci11081041

**Published:** 2021-08-05

**Authors:** Sami Obaid, François Rheault, Manon Edde, Guido I. Guberman, Etienne St-Onge, Jasmeen Sidhu, Alain Bouthillier, Alessandro Daducci, Jimmy Ghaziri, Michel W. Bojanowski, Dang K. Nguyen, Maxime Descoteaux

**Affiliations:** 1Departement of Neurosciences, Université de Montréal, Montreal, QC H3C 3J7, Canada; d.nguyen@umontreal.ca; 2Centre de Recherche du CHUM (CRCHUM), Montreal, QC H2X 0A9, Canada; jimmyghaziri@gmail.com; 3Centre Hospitalier de l’Université de Montréal (CHUM), Division of Neurosurgery, Montreal, QC H2X 3E4, Canada; alain.bouthillier@umontreal.ca (A.B.); m_bojanowski@yahoo.com (M.W.B.); 4Sherbrooke Connectivity Imaging Lab (SCIL), Sherbrooke University, Sherbrooke, QC J1K 0A5, Canada; francois.m.rheault@usherbrooke.ca (F.R.); eddemanon@gmail.com (M.E.); guido.guberman@mail.mcgill.ca (G.I.G.); etienne.st-onge@usherbrooke.ca (E.S.-O.); jsidhu@ubishops.ca (J.S.); maxime.descoteaux@gmail.com (M.D.); 5Department of Neurology and Neurosurgery, Faculty of Medicine, McGill University, Montreal, QC H3A 2B4, Canada; 6Department of Computer Science, University of Verona, 37134 Verona, Italy; alessandro.daducci@univr.it; 7Centre Hospitalier de l’Université de Montréal (CHUM), Division of Neurology, Montreal, QC H2X 3E4, Canada

**Keywords:** epilepsy, insula, operculum, connectome, diffusion magnetic resonance imaging, tractography

## Abstract

Operculo-insular epilepsy (OIE) is an under-recognized condition that can mimic temporal and extratemporal epilepsies. Previous studies have revealed structural connectivity changes in the epileptic network of focal epilepsy. However, most reports use the debated streamline-count to quantify ‘connectivity strength’ and rely on standard tracking algorithms. We propose a sophisticated cutting-edge method that is robust to crossing fibers, optimizes cortical coverage, and assigns an accurate microstructure-reflecting quantitative conectivity marker, namely the COMMIT (Convex Optimization Modeling for Microstructure Informed Tractography)-weight. Using our pipeline, we report the connectivity alterations in OIE. COMMIT-weighted matrices were created in all participants (nine patients with OIE, eight patients with temporal lobe epilepsy (TLE), and 22 healthy controls (HC)). In the OIE group, widespread increases in ‘connectivity strength’ were observed bilaterally. In OIE patients, ‘hyperconnections’ were observed between the insula and the pregenual cingulate gyrus (OIE group vs. HC group) and between insular subregions (OIE vs. TLE). Graph theoretic analyses revealed higher connectivity within insular subregions of OIE patients (OIE vs. TLE). We reveal, for the first time, the structural connectivity distribution in OIE. The observed pattern of connectivity in OIE likely reflects a diffuse epileptic network incorporating insular-connected regions and may represent a structural signature and diagnostic biomarker.

## 1. Introduction

The insula is a multimodal area involved in sensorimotor, autonomic, cognitive, and socio-emotional functions [1]. Previous functional and structural imaging studies have elucidated the extensive circuitry linking the insula to the surrounding frontal, temporal, and parietal lobes as well as subcortical structures [2,3,4,5,6,7,8]. The various roles of the insula and the ample distribution of the insular connectome may explain the diverse seizure manifestations observed in patients with operculo-insular epilepsy (OIE) that can include symptoms and signs reminiscent of frontal, parietal, or temporal lobe seizures [1,9]. Because clinical identification of OIE is challenging, non-invasive electrophysiological and imaging investigations including scalp electroencephalography (EEG) monitoring, magnetic resonance imaging (MRI), single-photon emission computed tomography (SPECT), positron emission tomography, and magnetoencephalography (MEG) [10,11,12,13,14,15,16,17] are typically warranted to support the diagnosis of insular epilepsy. However, such diagnostic tools are often limited in their ability to precisely localize the insula as the zone of seizure onset, ultimately requiring an intracranial electroencephalography (icEEG) study to accurately identify the epileptogenic zone [18]. It is likely that OIE remains an under-recognized condition, and additional non-invasive methods that would improve its recognition would be a welcome addition to the current diagnostic tools. In this regard, we previously showed that cortical thickness analysis may potentially help as patients with OIE exhibited widespread thinning of the ipsilateral insula and specific extra-insular areas connected to the insula [19].

Recent studies have shown that diffusion-weighted imaging (DWI)-derived tractography may be useful in the study of focal epilepsy, notably by preventing complications of epilepsy surgery, assessing the long-term consequences of chronic seizures, and even help distinguish between different types of focal epilepsies [20,21,22,23,24,25,26,27]. Indeed, studies have shown differential structural brain connectivity patterns in epileptic patients when compared to nonepileptic controls and also between patients with temporal and extratemporal epilepsies [21,22,23,24,25,26,27]. Focal epilepsy shows variations in ‘connectivity strength′ (CS) within the epileptic focus and in regions involved in the early spread of seizures [21,22,23,24,25,27,28].

Herein, we sought to assess, for the first time, the tractography-derived connectivity pattern in patients with OIE and evaluated if such changes revealed a characteristic and potentially specific distribution of CS alterations within the insular epileptic network. While most structural studies have employed standard probabilistic/deterministic or DTI tracking and quantified the CS using the debated streamline-count [22,23,24,27], we instead implemented a cutting-edge pipeline using surface-enhanced tractography (SET) [29] and Convex Optimization Modeling for Micro-structure Informed Tractography (COMMIT) [30,31] to address some critical limitations of tractography and compute quantitative microstructure-reflecting measures of connectivity. Our method is robust to crossing fibers due to the use of fiber orientation distribution functions (fODF) [32,33], optimizes coverage of the cortex and hard-to-track regions due to SET [29], and allows assigning a quantitative weight per connection thanks to COMMIT [30]. The COMMIT weight represents the intra-axonal cross-sectional area of the actual axonal fibers linking two anatomical areas [30] and therefore provides a more biological marker of CS.

## 2. Materials and Methods

### 2.1. Participants

We studied nine patients with long-standing refractory OIE (seven females; 30 ± 8 years; 18–44 years; five right OIEs and four left OIEs) treated at the University of Montreal Health Center. The epileptic focus involved a portion or the whole insula in all patients as well as the adjacent operculum (frontal, temporal or parietal) in eight patients. We also studied two age- and sex-matched control groups composed of 22 healthy individuals with no neurological or psychiatric disorders (10 females: 29 ± 5 years; 24–40 years) and eight patients with medically intractable TLE (four females; 27± 4 years; 20–34 years). Every epileptic patient underwent a standardized comprehensive evaluation including a detailed neurological history and examination, review of medical records, scalp-EEG video recordings of ictal seizures and a complete neuropsychological evaluation. Imaging investigations were performed in every epileptic patient and included seizure protocol T1, T2, and Fluid-attenuated inversion recovery (FLAIR) brain MRI sequences as well as an ictal SPECT. In addition, MEG was performed in six OIE patients and three TLE patients to better delineate the seizure focus. Furthermore, an icEEG recording was performed in eight OIE and one TLE participants. In order to specifically study OIE, we only selected patients who had a favorable outcome following a partial or radical insular resection with or without an operculectomy [34,35] (Engel class I for seven patients and II for two patients; mean follow-up time 4.2 ± 1.3 years). Similarly, to ensure confident focus localization, all patients in the TLE group had to have unilateral hippocampal sclerosis and good outcome after anterior temporal lobectomy (Engel class I at last follow-up; mean follow-up time 3.1 ± 1.6 years). A favorable post-operative seizure outcome allowed us to be certain about the localization of the seizure focus for all patients included in this study. All investigations including both noninvasive tests and icEEG studies were performed less than a year prior to surgical resection of the epileptogenic zone. Patients with tumoral lesions or vascular anomalies were excluded from the study. All healthy controls (HC) were scanned using the same MRI sequences as the epileptic patients.

### 2.2. Standard Protocol Approvals and Patient Consents

The study was approved by the University of Montreal Health Center ethics board and conformed to the Declaration of Helsinki. Informed consent was obtained from all participants.

### 2.3. Image Acquisition

All participants underwent the same acquisition protocol consisting of T1-weighted and DWI-weighted sequences on a 3T Achieva X MRI (Philips, the Netherlands). T1-weighted MRI data were acquired with the following parameters: TR = 8.1 ms; TE = 3.8 ms; flip angle = 8°; voxel size = 1 × 1 × 1 mm; FOV = 230 × 230 mm. The diffusion images were acquired with the following parameters: TR = 7.96 ms; TE = 77 ms; flip angle = 90°; voxel size = 1.8 × 1.8 × 1.8 mm; FOV = 230 mm. Diffusion-weighted images consisted of one pure T2-weighted image at *b* = 0 s/mm^2^ image and 60 images with noncollinear diffusion gradients at a *b* = 1500 s/mm^2^.

### 2.4. Image Processing and Connectivity Matrix Construction

Following DICOM-to-NifTI conversion of all acquired images [36], we launched Tractoflow version 2.2.0 (Appendix A: the code is available online at https://github.com/scilus/tractoflow/tree/2.2.0 (accessed on 5 August 2021)) [37], a recently published robust and efficient fully automatic tractography processing pipeline. Tracking maps obtained using Tractoflow consist of inclusion and exclusion probabilistic volume estimation (PVE) maps defining anatomically-constrained stopping criteria based on the T1 intensity of individual voxels [37,38]. The output of Tractoflow was then further processed through advanced steps to create the structural connectomes using SCILPY library version 1.0.0 (Appendix A: the library is available online at https://github.com/scilus/scilpy/tree/1.0.0 (accessed on 5 August 2021)). Probabilistic streamline tracking was launched using the surface-enhanced particle filtering tractography algorithm version 1.1 (Appendix A: the code is available online at https://github.com/StongeEtienne/set-nf/tree/v1.1.a (accessed on 5 August 2021)) [29,38] computed from constrained spherical deconvolution-derived fODFs [32,33] and the CIVET surface [39]. SET recreates, within the gyri′ white matter, a surface from which the tracking is initiated and terminated. The depth (surface flow) of the recreated surface from the gray/white matter interface is related to the number of iterations chosen, with a larger number of iterations leading to a deeper surface. Ten million streamlines were seeded at a surface flow of 100 iterations. Streamlines were excluded from the tractogram if their length was not within the 5–200 mm range, or if they exhibited significant looping (>330°). SET was chosen due to its ability to improve cortical coverage and improve the robustness of connectome building [29].

COMMIT was then used to filter the raw tractogram and compute COMMIT weights of individual streamlines [30]. The Freesurfer (available online: http://surfer.nmr.mgh.harvard.edu/ (accessed on 5 August 2021), RRID:SCR_001847) output computed from native T1 images was used to generate 246 cortical/subcortical regions of interest according to the Brainnetome anatomical atlas [40], to which three additional parcels were added (brainstem = 247, left cerebellum = 248, right cerebellum = 249). The COMMIT-weighted tractogram and Brainnetome parcellations were used to derive COMMIT-weighted structural connectivity matrices. Briefly, the COMMIT weight of a streamline is a measure that quantifies the actual contribution to the diffusion MRI signal of each individual streamline, and is proportional to the cross-sectional area of the biological fibers along their path [30]. The COMMIT weight of a connection corresponds to the sum of the individual weights assigned by COMMIT to each streamline connecting two parcels of the matrix, and was used as a marker of CS in our study. Through its ability to take into account the tracking bias related to variations in bundle width, the COMMIT weight constitutes a more biological proxy than the frequently used, but debated, streamline count [41,42]. Matrices of patients with right-sided OIE or TLE were side-flipped, which allowed the analysis to be performed uniformly. Corresponding bundles (connections) that were anatomically dissimilar (high shape variability) between HCs were excluded from the final matrices. To evaluate bundle similarity between HCs, we used a metric computed from the bundles’ binary masks registered in the MNI space. The metric calculates, for every HC, the minimal distance between each non-zero voxel contributing to a specific bundle in a specific HC from the nearest non-zero voxel of the average bundle of HCs. In each HC, the value is obtained by computing the average of the minimal distances for that specific bundle [43]. All bundles with an average minimal distance of more than 4 mm in over 10% of HCs were excluded (masked) in all matrices of the HC, OIE, and TLE groups. This criterion allowed for the inclusion of anatomically reliable and replicable bundles (Figure 1).

### 2.5. Group Comparisons of COMMIT-Weighted Matrices

COMMIT-weighted 249 × 249 whole-brain matrices were computed. In addition, sub-networks consisting of (i) 6 × 243 matrices linking the six subinsular regions to all 243 extra-insular regions (insula-extrainsula subnetwork matrices) and (ii) 6 × 6 matrices linking the six subinsular regions to each other (insular subnetwork matrices) were built. The similarity mask was created for the 249 × 249 whole-brain network and then recalculated based on (i) the 6 × 243 insula-extrainsula subnetwork and (ii) the 6 × 6 insular subnetwork. COMMIT-weighted matrices and submatrices were then compared in a group analysis using general linear models between (a) patients with OIEs and HCs, and (b) patients with OIE and patients with TLE. Age and gender were added as covariates for the OIE vs. HC comparison whereas age, gender, the age at onset of epilepsy, the duration of epilepsy, and the side of epileptic focus were included as covariates in the OIE vs. TLE analysis. Limited evidence suggests a vast but poorly characterized epileptic network in OIE [14]. Hence, given a lack of strong hypotheses at the level of the whole-brain network, we performed exploratory analyses for the 249 × 249 matrices. Analyses of subnetworks were then performed using a confirmatory approach. Between-groups difference maps of COMMIT weights were created using a threshold of *p* < 0.001 for exploratory analyses of whole-brain matrices and a threshold of false discovery rate (FDR)-corrected *p* < 0.05 for confirmatory analyses of both 6 × 243 insula-extrainsula and 6 × 6 insular subnetwork matrices. All group comparisons were implemented using FSL′s *randomize* algorithm (available online: http://www.fmrib.ox.ac.uk/fsl/ (accessed on 5 August 2021), RRID:SCR_002823) [44], which used 1000 permutations to build null distributions of group differences for each connection.

### 2.6. Group Comparisons of Graph Theoretic Measures

Structural networks can be characterized using graph theory measures, which allow a quantitative analysis of network topological properties that can be used to compare the structural organization of various pathologies including focal epilepsy [23,25,28,45,46,47,48,49]. Graph measures of each individual whole-brain matrix from all three groups were computed using the Graph Analysis Toolbox (GAT) version 1.5 [50] on MATLAB, version 18.0 (available online: http://www.mathworks.com/products/matlab/ (accessed on 5 August 2021), RRID:SCR_001622). Using GAT, we built undirected binary adjacency matrices in which any connection with a non-zero COMMIT weight was included in the network. We analyzed regional network measures, calculated for each node including (1) degree (number of connections to the node), (2) betweenness centrality (number of shortest paths that pass through a node), (3) clustering (fraction of connected triangles around a node), and (4) local efficiency (average of the inverse shortest path length in the neighborhood a node; correlates with clustering). We also assessed the following global network measures: (1) average degree, (2) average betweenness centrality, (3) average clustering coefficient, (4) characteristic path length (average of the shortest path length across all nodes), (5) global efficiency (average inverse shortest path length), and (6) small-worldness (ratio of average clustering coefficient to characteristic path length) [23,45,46,47]. The covariates added in connectivity matrices analyses were regressed out from graph theoretic measures. Adjusted measures of whole-brain networks were then compared using two-sample t-tests in GAT, and null distributions of group differences were created using 2000 permutations. Between-group comparisons were performed using a threshold of FDR-corrected *p* < 0.05 for regional measures and an uncorrected threshold of *p* < 0.05 for global measures.

### 2.7. Visualization

Three dimensional projections of structural connections and nodes were visualized using BrainNet Viewer (available online: https://www.nitrc.org/projects/bnv/ (accessed on 5 August 2021), RRID:SCR_009446) [51] for both comparisons of COMMIT weight matrices and graph theory analyses. The left side of the illustrated brains correspond to the side of seizure focus.

## 3. Results

### 3.1. Patient Population

Out of the nine patients with OIE, only three patients exhibited a small focal cortical dysplasia within the operculo-insula region on MRI. For that reason, the location of the epileptic focus relied on icEEG monitoring in all patients. Mesiotemporal sclerosis was absent in all OIE participants, but present in all TLE participants. Demographic and clinical data were similar between the OIE, TLE, and HC groups (Table 1). Analyses of COMMIT-weighted matrices and graph theoretic measures were therefore undertaken using comparable matched groups.

### 3.2. Group Comparisons of COMMIT-Weighted Matrices

#### 3.2.1. Comparison of the Whole-Brain Network

The average matrices of COMMIT weights for the OIE, TLE, and HC groups are illustrated in the Appendix A, respectively. Statistical comparisons revealed significant increases in COMMIT weights bilaterally in multiple bundles of OIE patients compared to HCs. A pattern of decreased COMMIT weights was also observed but was more limited (Figure 2; Appendix A). Similarly, a wider pattern of increased connectivity was detected in OIE patients compared to TLE patients, both ipsilateral and contralateral to the seizure focus (Figure 3; Appendix A).

#### 3.2.2. Comparison of Subnetworks

When isolating the insula-extrainsula subnetwork, significant increases in COMMIT weight were noted on the side of seizure focus between the dorsal granular insula and the pregenual cingulate gyrus in patients with OIE compared to HCs (*p*_FDR_ < 0.05). However, no variations were observed when comparing the insula-extrainsula subnetwork of patients with OIE to the same network of patients with TLE.

The average insular subnetwork matrices and connectome rings for all three groups are illustrated in Figure 4 and Appendix A. Group comparisons revealed a statistically significant increase in COMMIT weight between the dorsal agranular and dorsal granular insula ipsilateral to seizure focus when comparing OIE to TLE patients (*p*_FDR_ < 0.05; Figure 5). However, no differences were noted when contrasting the insular subnetworks of HCs and OIE patients.

### 3.3. Group Comparisons of Graph Theoretic Measures

#### 3.3.1. Regional Graphical Properties

Statistically significant regional differences in graph theoretic measures were observed when comparing patients with OIE to patients with TLE (uncorrected threshold of *p* < 0.05; Figure 6). A pattern of significantly increased degree and betweenness centrality was observed bilaterally in the OIE group and was more diffuse than in the TLE group. Interestingly, we observed higher values for both metrics within insular subregions of OIE patients ipsilateral to seizure focus. In contrast, the clustering coefficient and local efficiency was significantly elevated within the ipsilateral mesiotemporal subregions of patients with TLE. These findings, which are summarized in the Appendix A, did not survive FDR correction for multiple comparisons. Surprisingly, no significant differences were observed when comparing the OIE group to HCs.

#### 3.3.2. Global Graphical Properties

There were significant differences in global measures between patients with OIE and TLE (*p* < 0.05). Patients with OIE exhibited an overall higher average degree and global efficiency, but a lower average clustering coefficient and characteristic path length than patients with TLE. There was no between-group difference in average betweenness centrality or small-worldness. When comparing the OIE and HC groups, no differences in whole-network properties were found.

## 4. Discussion

Network neuroscience has gained significant popularity in the past two decades, particularly in the field of epilepsy [23,25,28,52]. The shift in our understanding of focal epilepsy from a focal disease to a localized circuitry [28,53] justifies the study of pathological networks rather than circumscribed foci. We therefore sought to evaluate, using refined methodological tools computing DWI tractography-derived networks, the pattern of structural connectivity alterations in patients with medically refractory OIE. We observed a wider pattern of hyperconnected regions in patients with OIE compared to patients with TLE or HCs. Subnetwork analyses revealed ‘hyperconnections′ between insular-connected regions in patients with OIE. In addition, when comparing the OIE to the TLE group, OIE patients exhibited an overall more efficient network on global graph theory assessment and disclosed various nodal alterations.

An ample distribution of hyperconnected areas was observed on whole-brain analysis in patients with OIE, both ipsilaterally and contralaterally. A vast pattern of bilateral increased CS was detected involving the frontal, parietal, occipital, temporal and insular cortices, the ipsilateral putamen, the ispsilateral thalamus, and the brainstem while a more limited pattern of decreased CS was found. Interestingly, these changes seem to reflect the diffuse cortical and subcortical connectivity of the insula [2,3,4,5,6,7,8] and are consistent with the vast operculo-insular epileptic network described in functional studies [14]. Moreover, a wider pattern of hyperconnectivity was observed in patients with OIE when compared to patients with TLE. In this regard, previous structural connectivity studies have revealed the pattern of white matter bundle alterations in TLE [22,24,27,54,55]. Although the results varied between studies, a pattern of increased connectivity involving the ipsilateral mesiotemporal region and associated limbic structures was commonly observed [22,24,54]. The limited connections of the mesiotemporal structures mainly incorporating, but not restricted to, the frontocentral and temporolimbic areas [19,56,57] likely constrain a more confined epileptic network in patients with TLE [22,24,27,54,55]. It is therefore not surprising that we observed a more extensive pattern of increased CS in patients with OIE. Furthermore, the FDR-corrected analysis of the insula-connected subnetworks revealed, in OIE patients, an ipsilateral increase in CS between the dorsal granular insula and the pregenual (anterior) cingulate gyrus. Comparing patients with OIE and TLE also disclosed an FDR-corrected increase in CS between the ipsilateral dorsal granular and agranular subregions. In this sense, the epileptic network of OIE was previously shown to involve subregions of the insula and regions heavily connected to the insula such as the mid to anterior cingulate gyrus [14,58,59]. It could be argued that propagation of epileptic discharges originating from the mesiotemporal structures may include insular subregions and result in connectivity alterations within the insular network of patients with TLE. However, as shown in our study, cases of pure OIE would be expected to result in more severe insular connectivity changes than patients with TLE.

The analysis of graph theoretical measures was also used to compare whole-brain networks of patients with OIE and patients with TLE and HCs. Graph theory analysis is a mathematical tool that allows the quantitative assessment of various types of networks [47] and has direct applications in the diagnosis and management of focal epilepsy [23,25,28,46,49,52]. When contrasting OIE to TLE patients, we observed that patients with OIE exhibited significantly higher values of degree and betweenness centrality within ipsilateral insular subregions while TLE patients disclosed increased clustering and local efficiency within ipsilateral mesiotemporal regions. In regional graph theory analysis, the large number of regions of interests, edges, and graphical measures may lead to restrictive thresholds and an exaggeration of type II statistical error following correction for multiple comparisons [23]. Hence, despite the lack of survival following FDR correction, the observed pattern of statistically significant regional alterations in the insular area of OIE patients and in the medial temporal area of TLE patients may suggest an increase in connectivity within these pathological regions. Similarly, studies evaluating the tractography-computed structural connectome of patients with TLE revealed ipsilateral hyperconnectivity within the ipsilateral epileptic network as shown by either increased local efficiency or increased clustering and degree [23,60]. It is therefore reasonable to believe that the observed intrainsular and mesiotemporal hyperconnectivity may be linked to the operculo-insular and temporal lobe epileptic network, respectively. Analysis of global measures between OIE and TLE patients unveiled, in the current study, an overall more globally efficient network (lower characteristic path length), a higher average degree, and a decreased average clustering coefficient in patients with OIE. Global efficiency is a measure of integration that characterizes the ease of information flow between regions, while the average degree and average clustering are related to the overall extent of connectivity and segregation within a network [23,45,46,47,61]. A previous meta-analysis incorporating both structural and connectivity studies revealed that patients with focal epilepsy typically display higher characteristic path lengths (lower global efficiency) and average clustering coefficients, which denotes an increase in segregation at the expense of a decrease in integration [25]. In other words, focal epilepsy leads to a more regular network organization, and such regularization seems to become more evident during the ictal phase of seizures [62,63]. As patients in both OIE and TLE groups exhibited focal epilepsy, the higher global efficiency in the OIE group may be explained by a less efficient TLE network resulting from disruptions of more key regional hubs and concomitant increased path length. Alternatively, as hyperconnected hubs may play a role in the pathophysiology of focal epilepsy [23], it is conceivable that patients with OIE disclosed more highly connected regions. This is further supported by the higher average degree observed in that population. On the other hand, the somewhat paradoxical lower average clustering coefficient observed in patients with OIE may be related to a reduction in connectivity [55]. In this regard, structural ‘hypoconnections′ have been linked to a decrease in average clustering in patients with focal epilepsy [55] and, although the pattern of increased connectivity was more widespread, the distribution of decreased CS observed on whole-brain analysis in patients with OIE was relatively vast (Figure 1).

Surprisingly, both regional and global graph properties were not different between patients with OIE and HCs. These findings were rather unexpected, especially considering the differential pattern of CS observed on whole-brain analysis. Two main factors could have potentially contributed to the absence of contrast between both groups. First, the threshold applied for the creation of binary adjacency matrix was such that any connection with a non-zero COMMIT weight was included. As such, matrices of HCs and OIE patients might have exhibited a similar binarized distribution. This suggests that the difference in the distribution of CS between both groups was likely due to a different weight of non-null connections rather than a differential pattern characterized by the presence of connections in one group and the absence of corresponding connections in the other group (i.e., binarization thresholded at a COMMIT weight of zero). The observed distinctive pattern between OIE and TLE patients likely reflects stronger differences between both groups that involve variations not only in the strength of non-null connections, but also in the distribution of binarized connections. Binarizing the adjacency matrix using a threshold of density [23,50] or simply using a higher COMMIT weight inclusion threshold may have selected connections with higher CS and led to distinctive graph theory patterns. Second, we applied the same similarity mask to both HC and OIE matrices. Although filtering is useful to preserve only anatomically reliable connections, it may have hidden the differential pattern between OIEs and HCs.

The pathophysiology of CS alterations in patients with focal epilepsy remains debated. Network and subnetworks analyses of CS revealed a pattern of increased connectivity in regions believed to be part of the operculo-insular epileptic network. We also found some regions of decreased connectivity on the whole-brain assessment. Besson et al. have previously shown, in patients with focal epilepsy, a distribution of decreased DWI-derived bundle density in connections linking epileptogenic zones to non-epileptogenic zones and linking non-epileptogenic zones to each other [52]. Interestingly, regions of preserved connectivity were also observed between epileptogenic zones and between epileptogenic and propagation zones. They suggested that the maintained structural links were probably related to pathological hyperconnected regions at the expense of decreased distant connectivity within non-epileptogenic regions [52]. The increased connectivity may sustain seizure organization and propagation while the distant decreased connectivity may result from abnormal plastic changes beyond the impact of seizure [52] or may be related to deafferentation of connections originating from the epileptogenic zone [64]. In an earlier report by Bonilha et al., a decrease in connectivity in limbic regions of patients with TLE was associated with a paradoxical increase in clustering, local efficiency, degree, and betweenness centrality [23]. Such modifications may be due to a sequence characterized by seizure-induced axonal loss, development of aberrant clustered connections, and reorganization of limbic networks [53]. Ultimately, this may lead to self-reinforcing excitation and facilitation of epileptogenicity [23]. While the exact pathomechanism of hyperconnectivity within epileptogenic zones is still unknown, previous studies in focal epilepsy have suggested that it may be related to increased adaptative axonal sprouting [65,66] or even neurogenesis [67]. These abnormal networks may in turn generate abnormal synchronous epileptic bursts [68]. Even though these hypotheses originate from studies in extra-insular epilepsy, the rationale stems from dysfunctional circuitry in focal epilepsy and can therefore be applied to OIE. It is reasonable to believe that the observed connectivity changes in patients with OIE may result from similar pathological processes affecting a particularly vast insular network. Furthermore, we previously described the pattern of cortical thinning in patients with OIE [19]. Interestingly, the ample distribution of cortical thinning in OIE seems to mirror the extensive pattern of connectivity alterations [19]. Cortical atrophy in patients with OIE probably reflects a pathological process related to the operculo-insular epileptic network, resulting in glutamatergic excitotoxicity and concomitant neuronal death within a vastly hyperconnected network or, alternatively, deafferentation of connections constituting the more limited network of decreased connectivity [19].

Just like other types of focal epilepsies, OIE constitutes a disease characterized by altered connections within a network. The use of *quantitative* structural connectivity measures is therefore essential to accurately define the epileptic network. To do so, cutting-edge quantitative connectivity tools using novel technologies are of paramount importance. In our study, the use of state-of-the-art methods allowed us to ascertain the building of reliable connectivity matrices and to compute trustworthy measures of CS. Using our elaborate pipeline, we were able to evaluate the pattern of structural connectivity in patients with OIE. SET is a newly developed tracking algorithm that optimizes targeting of the cortical surface in difficult-to-track regions and has the ability to counter the known gyral bias of standard tractography algorithms, leading to connectivity matrices with denser and more reproducible populations of fibers [29]. The use of probabilistic tractography also enabled a better depiction of fiber curvatures and fanning, therefore mitigating network scattering by providing a comprehensive view of connections [24]. We also used COMMIT weights as a measure of CS. Most studies of structural connectivity alterations in focal epilepsy use the debated streamline-count to quantify the intensity of connections [22,24,27,69]. However, it was previously shown that, even using the most sophisticated tracking algorithms, the number of reconstructed streamlines is influenced by the length, curvature, degree of branching [41], and width of white matter tracts [38] and that streamline-count cannot be used as a quantitative diffusion MRI marker. Even with the addition of the fODF model to characterize crossing fibers, accurate differentiation between intersecting, kissing, and branching patterns is impossible [41,70]. Counting streamlines may therefore not constitute an adequate measurement of CS [41]. While we recognize that COMMIT cannot overcome all the limitations related to the morphology of white matter tracts, bundle width can be addressed [30]. By comparing the estimated streamlines to the initial diffusion MRI signal measured both locally and globally, COMMIT assigns a quantitative normalized weight to individual streamlines [30]. It has the ability to increase the weight of streamlines within the commonly under-represented small bundles and decrease the weight of streamlines making up the typically over-displayed large tracts. By doing so, the effective contribution of each tract is recovered. For those reasons, the weight assigned by COMMIT to a connection may represent a more accurate and biologically interpretable metric of CS. In addition, COMMIT allows the removal of unexplainable streamlines (Daducci et al., 2015), leading to a reduction in the rate of false positive connections [30,42] and concomitant improvement in the calculation of graph theory measures [71]. Furthermore, we calculated a mask of similarity in our HCs and applied it to all three groups. We chose to compute the mask based on the tractogram of HCs in order to exclude bundles that were morphologically dissimilar between members of a normal healthy population, therefore only selecting anatomically replicable and likely existing tracts. Similarity filtering attempted to remove false-positive tracts inherent to bundle tracking [42] and favored the building of reliable connectomes. Moreover, our connectivity matrices were built using an atlas with precise fine-grained parcellations [40] including subdivisions of the insula. This allowed us to study connectivity variations within insular subregions and more precisely characterize the epileptic network. Finally, our pipeline computed the connectivity matrices in the subjects′ native space, enabling accurate cortical mapping, and obviating the need for nonlinear deformations to a common space that may result in mismatching of tracks [62].

The present study provides unique information regarding the structural connectivity pattern in patients with OIE. It is nevertheless limited by a relatively small sample size, which is due to the rarity of OIE and the difficulty to establish its diagnosis. To address this limitation, patients with both lesional and non-lesional OIE were grouped together. Cortical dysplasia within the operculo-insular region was observed in three patients with OIE, which could have influenced seeding or targeting of insular-connected streamlines by blurring the white matter–gray matter interface. In addition, patients with OIE originating from the anterior insula tend to have a different epileptic network than patients with posterior OIE [14,34]. In our study, the epileptogenic zone may have involved different subregions of the insula. Analyzing subgroups independently could have potentially led to distinctive patterns of connectivity alterations. Furthermore, patients with right-sided OIE or TLE were side-flipped. Combining patients with both right- and left-sided epilepsies has been commonly performed in previous structural connectivity studies of patients with TLE [24,62,70] and constitutes an important step that allows the analysis to be performed uniformly. While it could be argued that patients with focal epilepsy can exhibit inter-hemispheric differences in connectivity alterations [22], the distribution of insular connectivity seems rather symmetrical [3,4,6,7] and it is therefore conceivable that homologous connections may exhibit similar ipsilateral changes in CS, regardless of the side of seizure onset. Despite these constraints, the rarity of OIE drove the pooling of all patients with the goal of improving statistical power and optimizing between-group comparisons.

In our analysis, we only included patients who became seizure-free following surgery of the operculo-insular region to make sure they really had OIE. Eventually, it would be interesting to accumulate enough patients with clear OIE and poor surgical outcome to assess if tractography-derived structural connectivity can be used to predict seizure outcome.

## 5. Conclusions

To our knowledge, our results reveal, for the first time, the alterations of structural connectivity in patients with OIE. To do so, we implemented a reliable pipeline based on robust tools that allowed us to quantitatively describe morphological connectivity changes in focal epilepsy. The wider pattern of increased CS observed in patients with OIE could suggest a more diffuse epileptic network than TLE. In addition, the ipsilateral increase in connectivity within insular subregions as well as between the insula and insular-connected regions likely reflects a densely connected insular epileptic network. Clearly, more work is necessary before we can consider the observed distribution of connectivity as a structural signature that could be used as a diagnostic biomarker of OIE. Given the small sample size and the difficult-to-prevent methodological limitations, these results must be considered preliminary and warrant further investigation through larger studies that would include other types of focal epilepsies.

## Figures and Tables

**Figure 1 brainsci-11-01041-f001:**
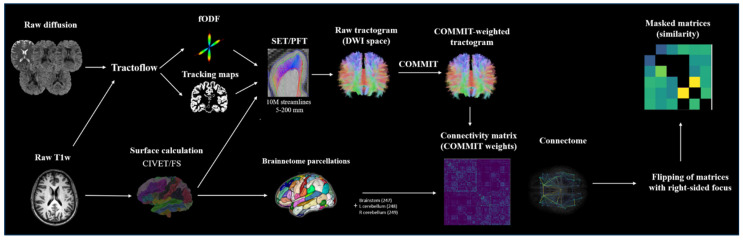
Processing flowchart. Raw images were processed using Tractoflow. The output of Tractoflow and CIVET-calculated surfaces were used to build the tractogram using SET, which was then processed with COMMIT. In parallel, the Freesurfer-calculated surfaces and segmentation were used to generate Brainnetome parcels. The COMMIT-weighted tractogram and Brainnetome parcellations were used to derive structural connectivity matrices. Matrices of patients with right-sided OIE or TLE were then side-flipped and bundles that were anatomically dissimilar between HCs were excluded in all matrices.

**Figure 2 brainsci-11-01041-f002:**
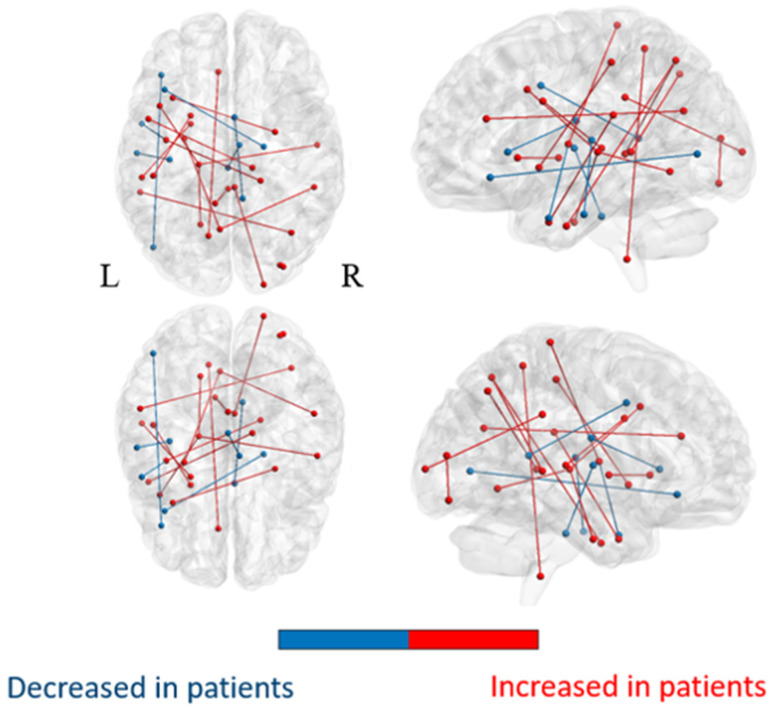
Illustration of a group comparison of whole-brain COMMIT weights between HCs and OIE patients. Significance was thresholded at *p* < 0.001 uncorrected. Matrices in both HCs and OIE patients were masked based on the similarity threshold calculated in HCs.

**Figure 3 brainsci-11-01041-f003:**
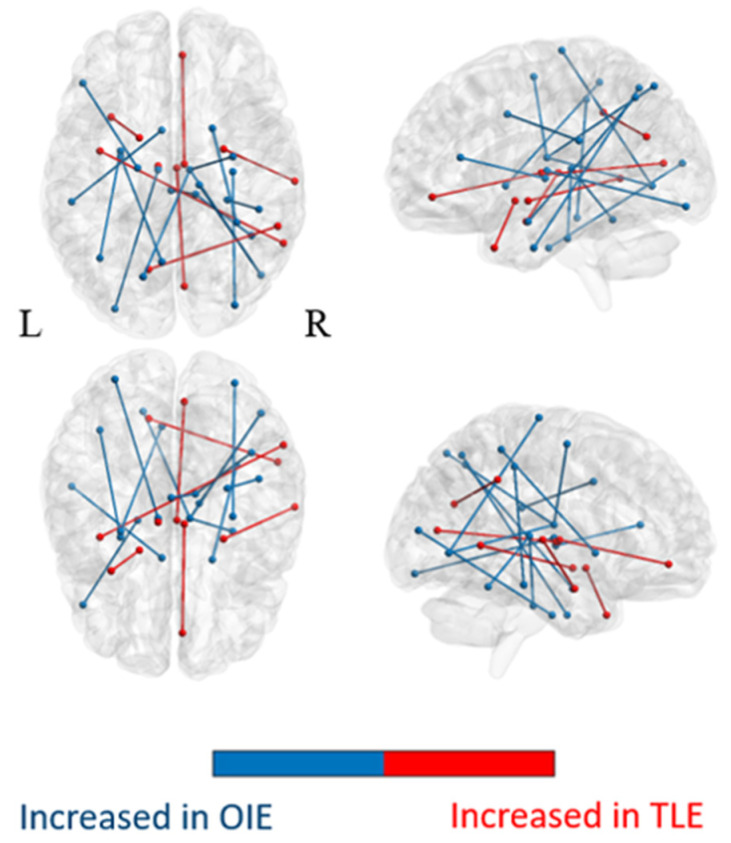
Illustration of a group comparison of whole-brain COMMIT weights between OIE and TLE patients. Significance was thresholded at *p* < 0.001 uncorrected. Matrices in both groups were masked based on the similarity threshold calculated in HCs.

**Figure 4 brainsci-11-01041-f004:**
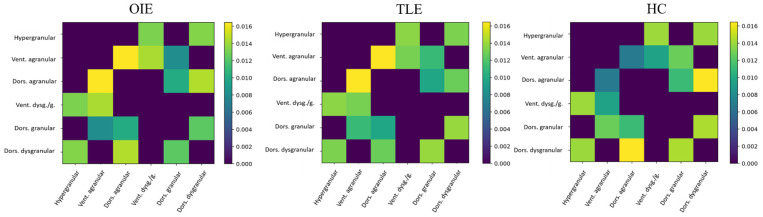
Average COMMIT-weighted connectivity matrices of the insular subnetwork in OIE, TLE, and HC participants. The submatrix was masked based on a similarity threshold calculated from the insular submatrices of HCs. The color bar represents the measured COMMIT weight. Vent. = ventral; Dors. = dorsal; dysg./g. = dysgranular/granular.

**Figure 5 brainsci-11-01041-f005:**
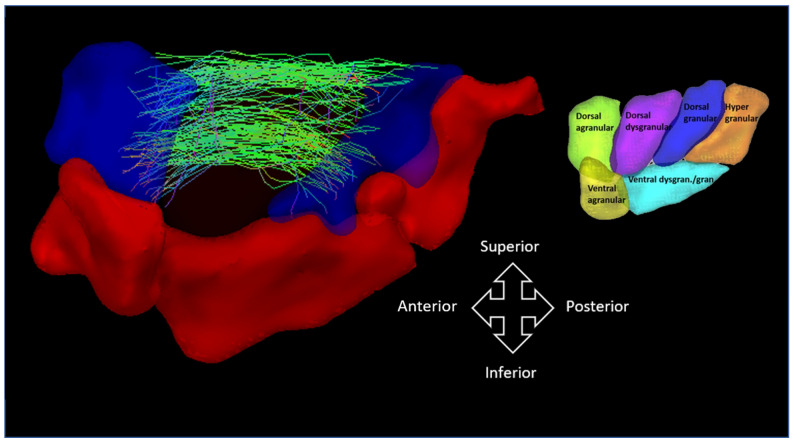
Illustration of a group comparison of COMMIT weights between of OIE and TLE patients when analyzing the insular subnetwork. An increase in CS was observed between the ipsilateral dorsal agranular (anterior blue region) and the dorsal granular insular subregion (posterior blue region). To better visualize the white matter bundle, the dorsal dysgranular subregion was voluntarily removed. The six Brainnetome atlas subregions of the left insula are illustrated on the right. Comparisons were performed using general linear models. Significance was thresholded at an FDR-corrected *p* < 0.05. Matrices in both groups were masked based on a similarity threshold calculated in HCs. gran. = granular; dysgran. = dysgranular.

**Figure 6 brainsci-11-01041-f006:**
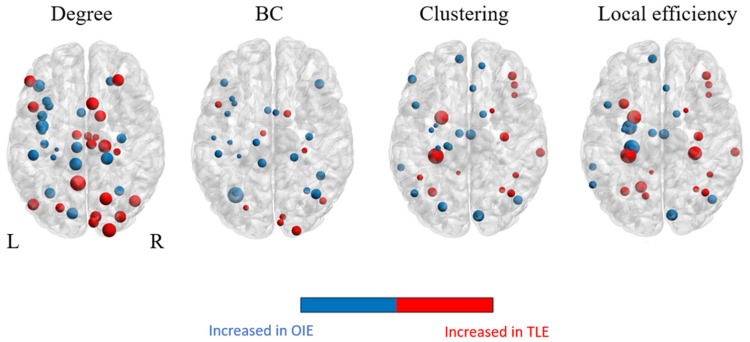
Illustration of a group analysis of regional graph theoretical measures when comparing OIE to TLE patients. The size of the colored nodes is related to the value of the metric; larger nodes correspond to higher values. Comparisons were performed using two-tailed t-tests. Significance was thresholded at *p* < 0.05 uncorrected. BC = betweenness centrality.

**Table 1 brainsci-11-01041-t001:** Demographic and clinical information.

	Age at MRI	Women	Age of Onset	Left-Sided Epilepsy	Duration
OIE (*n* = 9)	30 ± 8 (18–44)	7	16 ± 10	4	16 ± 12
TLE (*n* = 8)	27 ± 5 (20–34)	4	16.5 ± 10	5	11 ± 10
Healthy controls (*n* = 22)	29 ± 5 (24–40)	10	NA	NA	NA

Age at MRI, age of onset, and duration of epilepsy are shown in years ± SD (age range). The Mann–Whitney U test was performed for comparison of continuous variables (age at MRI, age of onset of and duration of epilepsy) while the chi-square test was used for categorical variables (gender and side of epilepsy). None of the between-group comparisons revealed statistically significant differences.

## Data Availability

The data presented in this study are available on request from the corresponding author. The data are not publicly available due to privacy restrictions.

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
