# Peer review of "Structural Connectivity Alterations in Operculo-Insular Epilepsy"

_brainsci, 2021, doi:10.3390/brainsci11081041_

Round 1

Reviewer 1 Report

Overall a good neuroimaging work in OIE and epilepsy network, however, there are some points to clarify and some questions/comments...

  • How was the diagnosis established in OIE? One patient of TLE received icEEG in TLE?
    • “In order to confirm the epileptogenic zone, an icEEG recording was performed in eight OIE and one TLE participants”
    • “In all patients of the TLE group, we ascertained the inclusion of patients with a seizure focus within the mesial temporal lobe through non-invasive investigations revealing hippocampal sclerosis and medial temporal lobe epilepsy.
  • What was the temporal association with the MRI, icEEG, surgery? 
  • Was there any association among the connectivity, EEG/icEEG and MEG, if analyzed?
  • Is there any patient excluded from OIE group because of the poor Engle class after surgery? If so it would be interesting to see the connectivity difference within the group and/or also validate the connectivity map of OIE. Any attempt were made to explore the favorable vs non-favorable OIE and validating the connectivity and cross reference to the clinical, imaging and electrophysiological concordance?
  • Authors present epileptic network in OIE with potential description of “signature” network that seems overstatement in absence control/negative control.

Reviewer 2 Report

The authors evaluated the structural connectivity distribution in operculo-insular epilepsy (OIE) using a novel method named  COMMIT (Convex Optimization Modeling for Microstructure Informed Tractography)-weight. The results regarding the distribution of connectivity may represent a potential biomarker in the diagnosis of OIE. 

The study is interesting, well described and the data are clear.

This is a very interesting paper.
